# Intralesional Treatments for Invasive Cutaneous Squamous Cell Carcinoma

**DOI:** 10.3390/cancers16010158

**Published:** 2023-12-28

**Authors:** Gloria Baeza-Hernández, Javier Cañueto

**Affiliations:** 1Department of Dermatology, Complejo Asistencial Universitario de Salamanca, Paseo San Vicente 58-182, 37007 Salamanca, Spain; gloria.baezahdez@outlook.com; 2Instituto de Biología Molecular y Celular del Cáncer (IBMCC), Centro de Investigación del Cáncer (CIC)-CSIC, Laboratory 20, 37008 Salamanca, Spain; 3Instituto de Investigación Biomédica de Salamanca (IBSAL), Complejo Asistencial Universitario de Salamanca, Paseo San Vicente 58-182, Hospital Virgen de la Vega, 10ª Planta, 37007 Salamanca, Spain

**Keywords:** CSCC, intralesional, methotrexate, 5FU, bleomycin, interferon, oncolytic viruses, cancer vaccines, intralesional immunotherapy

## Abstract

**Simple Summary:**

Cutaneous squamous cell carcinoma (cSCC) is the second most frequent cancer in humans, and it is especially common in fragile, elderly people. Surgery is the standard treatment for cSCC but intralesional treatments can be an alternative in those patients who are either not candidates or refuse to undergo surgery. Classic intralesional treatments, including methotrexate or 5-fluorouracil, have been implemented, but there is now a landscape of active research to incorporate intralesional immunotherapy and oncolytic virotherapy into the scene, which might change the way we deal with cSCC in the future. In this review, we focus on intralesional treatments for cSCC (including keratoacanthoma), from classic to very novel strategies.

**Abstract:**

Cutaneous squamous cell carcinoma (cSCC) is the second most frequent cancer in humans and has the potential to progress locally, metastasize, and cause death in a subset of patients. cSCC is especially common in the elderly, and it will probably represent a major health concern in the near future. Surgery is the standard treatment for cSCC, but intralesional therapies can sometimes be considered for certain patients and under certain circumstances. The choice of intralesional treatment depends on the patient′s characteristics and the clinician′s previous experience and expertise. Here we are reviewing intralesional treatments for cSCC and keratoacanthoma (KA). We have started with some classic drugs, such as methotrexate and 5-fluorouracil, bleomycin, interferon, and cryosurgery, but also comment on electrochemotherapy. Finally, we have focused on novel therapies, some of which are under development, and future perspectives, including intralesional immunotherapy and oncolytic viruses.

## 1. Intralesional Treatments for Invasive Cutaneous Squamous Cell Carcinoma

Cutaneous squamous cell carcinoma (cSCC) derives from the malignant proliferation of keratinocytes [1]. cSCC may arise de novo or from potential precancerous lesions such as actinic keratosis, actinic cheilitis, or chronic inflammatory diseases [2]. While it usually exhibits benign clinical behavior, cSCC has the potential to progress locally, metastasize, and cause death in a subset of patients [1]. Ten-year survival after surgery exceeds 90% for cSCC but drops dramatically when metastases occur [3]. The frequency of lymph node metastases is around 4%, and mortality rates are nearly 2%. Given its high frequency, cSCC has a significant impact on overall mortality [4]. It is responsible for the majority of deaths from skin cancer in people older than 85 years and, in some areas of the US, it has a mortality comparable to that of renal, oropharyngeal, and melanoma carcinomas [5]. cSCC is especially common in the head and neck area of males older than 65 years due to chronic ultraviolet light exposure, and as life expectancy increases, it will probably continue to represent a major health concern in the future [2]. 

Surgery is the standard treatment for cSCC, but sometimes intralesional therapy can be considered depending on clinical factors, such as the number of lesions, the size and the location of the tumor, and on patient features, such as immunosuppression, organ transplant reception, fragility, comorbidities, or a refusal to undergo surgery [2]. Intralesional treatments are administered directly inside the tumor, thus providing a selective drug delivery that prevents most systemic side effects [6]. This approach may be especially useful in patients with comorbidities and in those cases in which surgery is likely to result in either functional or cosmetic impairment [6].

In this review, we will focus on intralesional treatments for invasive cSCC, including keratoacanthoma (KA). There is some controversy on the nosologic position of keratoacanthoma and some authors have suggested it is an entity different from cSCC, while others claim KA is indeed a variant of cSCC. Notwithstanding that, the treatment strategies are similar and, in most papers on intralesional treatments, both cSCC and KA are considered together [7]. For that reason, we focus on both these entities. We have purposely excluded in situ squamous cell carcinoma from this review, as it will be covered elsewhere in this Special Issue. We will focus both on classical treatments and on future directions which will come in the upcoming years.

## 2. Classic Treatments

### 2.1. Methotrexate

Methotrexate (MTX) is a folate antagonist which binds to dihydrofolate reductase, inhibiting folate metabolism and thus RNA and DNA synthesis. Although there is no standardized treatment protocol, it has become the preferential intratumoral treatment in cSCC for some physicians. MTX was initially implemented in KA, but it has also been incorporated into the therapeutic arsenal of cSCC with good results, according to some studies.

Intralesional MTX (ilMTX) reports in KA are summarized in Table 1 [7,8]. Common doses vary between 12.5 and 25 mg with 1–2 week intervals. Many published papers are based either on case reports or small case series. Martorell-Calatayud et al. conducted a randomized clinical trial in which patients with histologically proven KA measuring >1.5 cm in diameter were treated either with ilMTX before surgery or not. The authors demonstrated a reduction from 50% to 80% of the lesion size in those cases managed with ilMTX before surgery compared with those which were not treated [9]. In a non-randomized prospective study, 73 clinical or histology-proven KAs were treated with ilMTX and the authors demonstrated a cure rate of 88%. In that paper, eight out of nine tumors which did not resolve with ilMTX decreased their size and were surgically excised without complications afterwards [10]. Treatment with ilMTX has been reported in combination with acitretin for the treatment of multiple KAs as well [11].

ilMTX has been used to treat cSCC as well (Table 2) [8]. In a retrospective comparative cohort study by Salido-Vallejo et al., 43 patients with histologically proven cSCC who were treated with neoadjuvant ilMTX showed a reduction in the tumor size [39]. For tumors ≥ 2 cm, there was a statistically significant difference in the number of surgical complex reconstructions between the group managed with neoadjuvant ilMTX and the group managed with surgery alone [39]. In a post hoc analysis of the group treated with ilMTX, they observed that the greatest reduction in tumor size occurred when tumors were located on the lower lip [39]. Indeed, in a prospective study by Bergón-Sendín et al. with 10 patients with cSCC, 9 of which were located the lower lip, all showed a decrease in size with neoadjuvant ilMTX and 30% showed complete response [40]. This reduction in tumoral size with ilMTX has been observed in other studies [41,42]. A prospective study of 40 patients from the same group showed that ilMTX reduced not only tumor diameter (area) but also tumor thickness when evaluated with ultrasonography [43].

A multicenter retrospective study by Gualdi et al. which included 21 patients with cSCC and 12 patients with KA treated with ilMTX showed a 92% complete response rate in KAs and 47.6% in cSCCs [38]. A total of 14.3% patients with cSCCs showed a partial response and were successfully treated with surgery afterwards [38]. This paper pinpointed towards a greater response in KA with ilMTX. Finally, a recent clinical trial (comparing ilMTX vs. intramuscular MTX for the treatment of KA, cSCC, and BCCs) showed a better (statistically significant) response in favor of the group treated with ilMTX, with fewer adverse events [47,48]. A total of 88.9% of the KAs completely resolved, but only 28.6% of the cSCCs completely resolved.

Regarding ilMTX safety, no serious adverse events have been reported in the majority of the cases or studies published [7,8,9,10,38,39,40,41,42,43,46]. The most common secondary effects are mild local pain or erythema [8,39,40,42]. However, special caution is advised when treating patients with renal failure as most studies exclude them, and in some cases, mucositis and pancytopenia have been reported in those patients [7,10,49]. In the study of Gualdi et al., one-third of the patients developed side effects, the most frequent being anemia (*n* = 5) and elevation of transaminases (*n* = 4), apparently not dose-dependent, and only 1 case of severe anemia required transfusion [38]. In one of the authors’ experiences, an elderly man developed systemic methotrexate toxicity (pancytopenia, mucosal ulcers) when treating a lower lip cSCC with 25 mg ilMTX (2 injections, 1 week apart), without previous kidney failure or contraindications for MTX treatment. He recovered with folinic acid treatment. Also, we have observed the aggressive evolution of a cSCC treated with ilMTX in a young female with Ferguson-Smith syndrome.

Most studies on the management of cSCC and KA with ilMTX lack long-term outcomes, as many of them just report a few months of follow-up [7,10,38,39,40,41,46]. It is likely assumed that cases that have achieved complete clearance using this treatment will continue with no recurrence as if they would have been managed with surgery and clear surgical margins. Also, the level of evidence mainly comes from series of cases and there is a need for more randomized control trials comparing dosage, treatment protocols, and long-term outcomes of patients.

### 2.2. 5-Fluorouracil

5-fluorouracil (5-FU) is an inhibitor of thymidylate synthetase, thus inhibiting DNA synthesis. Its intralesional use (il5FU) in KA and cSCC has been reported since the 1960s (Table 3 and Table 4) with good results [7,50]. Despite a lack of standardized protocols, it usually needs more treatments than ilMTX to achieve complete clearance. Usually, 0.2–2 mL of 5% 5-FU are injected weekly or every 3–4 weeks [7,50,51]. The largest case series of KAs treated with il5FU showed an efficacy of 97.5% (with 40 out of 41 KAs resolved) [52]. A very recent case series by Marka et al. showed dilute il5-FU cleared 39/40 KAs and 9/10 cSCCs [53]. In a retrospective study by Maxfield et al. that included 172 cSCCs (of which 7 were KAs), 92% of them resolved (158/172) and only 1 KA relapsed after il5-FU [54]. A phase 1 clinical trial (NCT03370406) is recruiting patients with cSCC in the lower extremities for treatment with il5-FU and topical imiquimod [55]. il5-FU is a well-tolerated procedure, with adverse events being mostly mild and local (pain and injection site reactions) and rarely systemic (including headache, dizziness, and nausea) [51,53,54,56,57]. Follow-up time has been inconsistent among the case reports and series, most of them less than 1 year, if stated [7,51]. il5-FU has been prescribed in combination with acitretin, 5-FU chemowraps, topical 5-FU, or imiquimod and after debulking [58,59,60,61,62,63]. Again, the level of evidence mainly comes from series of cases and there is a need for more randomized control trials comparing dosage, treatment protocols, and long-term outcomes of patients.

### 2.3. Bleomycin

Bleomycin (BL) is a glycopeptide antibiotic mainly used as an antineoplastic drug. It binds to metal ions which then produce reactive oxygen species, damaging the DNA. Old case reports show the efficacy of intralesional BL in KAs and cSCCs located mainly on the face [82,83,84,85]. Cumulative doses of 0.2 to 2.4 mg divided into 1 to 4 injections have been used in KAs, up to 2.4 mg in the case of keratoacanthoma centrifugum marginatum (24 injections total) and 115 mg of oil BL was used in a single case report of cSCC [7,82,83,84,85]. It seems to have a quicker effect than 5-FU with complete resolution of the lesions even after one injection [7]. Injection with local anesthetic is recommended as it is usually painful [7]. Side effects include local reaction and pigmentation, fever after the injection, and granulomatous inflammation in association with lipid material in the dermis (oleogranuloma) with the use of oil BL [7,83].

### 2.4. Interferon

Interferons are a group of cytokines involved in the upregulation of the immune system and especially involved in the response to viral infections and tumor suppression. Interferon alpha is a subgroup belonging to the type I interferon subclass, responsible for NK cell activation, proliferation of B cells, and CD8 T cell response enhancement [86]. Interferon-alpha 2a, with a molecular weight of 40 kDa, is administered in a fixed dosage, while interferon-alpha 2b, which is smaller with a molecular weight of 12 kDa, is administered with a dosage adjusted for body weight. These differences do not appear to affect their antigenicity as both bind to the same type I IFN receptor. Interferon-alpha 2b can be more familiar to dermatologists as it was used for adjuvant treatment in melanoma.

Intralesional interferon-alpha (ilIFNɑ) 2a or 2b has been scarcely reported for the treatment of KAs, especially on the face and scalp [7,87,88,89,90,91]. Doses varying from 8.1 to 117 MIU were used divided into up to 38 injections [7,87,88,89,90]. Two patients with recurrent cSCC on the face were treated with ilIFNɑ, with dosages of 2 MIU three times a week for 7 weeks and 1.5 MIU three times a week for 8 weeks and then biweekly for 3 months, respectively, with complete response after 6 and 8 years of follow-up [92]. The side effects included pain with injection and flu-like symptoms, so baseline and follow-up tests that include complete blood count, liver function, creatinine, and urea levels are recommended [7,87,88,89,90,91,92]. The evidence of ilIFN in the treatment of cSCC and KAs is scarce and only small case series have been reported, so there would be a need for further evidence.

### 2.5. Interleukin 2 (IL-2)

Interleukin 2 (IL-2) is a type of interleukin, a group of cytokines, which is produced by T cells and is involved in their proliferation and differentiation. It activates NK cells and induces B cell proliferation and antibody synthesis [93]. It has been used in advanced clear cell renal carcinoma and melanoma but its severe cardiovascular, respiratory, and infectious complications have limited its use. Combined intralesional IL-2 (8–10 MUI per session, weekly) and topical imiquimod have been used in a multifocal high-grade cSCC on the face occurring in a double-transplant patient (kidney and liver) with complete tumor clearance after 3 months of follow-up and no repercussions in liver or kidney function [94].

### 2.6. OK-432 (Picibanil)

OK-432 (picibanil) is a lyophilized preparation of Streptococcus pyogenes type A treated with benzylpenicillin potassium used as an immunotherapy anticancer drug and for ascites and pleural effusions and lymphatic malformations as well [95,96]. It is thought to stimulate the immune system with local inflammation [95]. Intralesional OK-432 (picibanil) has been used in an in-transit relapse of cSCC (2 clinical units every 2 weeks) with a complete response and no evidence of recurrence after 5 years [96]. Despite its apparent promising performance, its use has been anecdotical and it is difficult to provide recommendations based on this unique report.

### 2.7. Photodynamic Therapy

Photodynamic therapy combines light treatment with a photosensitizer (usually 5-aminolevulinic acid, methyl-aminolevulinate, or methylene blue) to selectively destroy precancerous and cancerous cells. A case report of cSCC on the cheek treated with intralesional photodynamic therapy has been published using 0.8 mL of 10% 5-aminolevulinic acid solution, which caused almost unbearable pain during the procedure [97]. After an occlusion period of 4 h, it was illuminated with red light (570–670 nm) using a Waldmann PDT 1200 (noncoherent light source) at a light dose of 100 J/cm^2^ and a fluence rate of 100 mW/cm^2^, with an intense burning sensation [97]. Local erythema and edema persisted for 5 days. A complete clinical response was achieved and after 16 months of follow-up and no recurrence was observed [97].

### 2.8. Cryosurgery

Resolution of cSCC with intralesional cryosurgery has been reported. Four patients with cSCC or KA located on the leg, hand, or face were treated with this method, one of them in combination with MTX. Under local anesthesia, an 18-gauge needle was inserted through the center of the tumor and connected to a liquid nitrogen gun, then treatment with two freeze (30–60 s with 5 mm free margins)-thaw cycles was performed [98]. For tumors larger than 15 mm, two needles were used perpendicularly [98]. All patients achieved complete response and no relapse after a follow-up of 1.4–9.5 years, with a good cosmetic result [98].

### 2.9. Electrochemotherapy

Electrochemotherapy (ECT) is a localized form of chemotherapy that combines electroporation and a chemotherapeutic agent, commonly intratumoral or intravenous bleomycin and, rarely, cisplatin, to achieve local tumor control [99,100]. When applying electric pulses to tumor tissues, cell membranes become permeable to the chosen drug, enhancing the local cytotoxicity of the anticancer agent [99]. The updated standard operating procedures for ECT of cutaneous tumors and skin metastases were published in 2018, and among its indications, primary skin cancers, including recurrent tumors and cutaneous metastases of any histology, are included [99]. It is a treatment procedure that should be considered for patients who refuse or are not good candidates for surgery [101,102,103]. Some authors recommend intravenous administration when there is extensive disease, lymphedema, or other conditions that could affect the distribution of the drug within the tumor, while intratumoral administration would be adequate for poorly perfused nodules [101].

ECT has been scarcely reported as a treatment for KA, including multiple lesions on the legs [104,105,106]. More data are available for the treatment of cSCC. The ESOPE prospective study included three patients with cSCC among other types of cancer, mainly melanoma, with limited side effects [107]. Intravenous injection of bleomycin was superior to intratumoral injection of bleomycin or cisplatin if the nodules treated were bigger than 0.5 cm^3^ [107]. In a recent prospective cohort (INSPECT 2008–2019), 156 patients with 284 cSCCs were treated with ECT with an overall response of 80% and a complete response in 63%, results significantly lower than for BCC and Kaposi sarcoma [100]. Patients with cSCC in this cohort showed an overall response rate of 83% with a favorable safety profile and data showed that intravenous bleomycin was superior to intratumoral administration and 1-year local progression-free survival was significantly better in patients with primary cSCC than in patients with locally advanced disease [108]. Another prospective study of 41 patients with cSCC showed an overall response of 85.2% (40.7% of CR) without serious systemic adverse events and lower grade ≥ 3 skin toxicity in patients with tumors smaller than 2 cm [109]. A similar overall response rate was observed in a retrospective study of 22 patients with advanced cSCC stage III T2N0M0 (seventh edition, American Joint Committee on Cancer staging system) with a very low complication rate [110].

In a prospective study of 39 patients with cutaneous and subcutaneous metastases, ECT was implemented as a palliative approach, and in 5 of those cases, cSCC had been the primary tumor [111]. No major adverse events were observed and ECT efficacy was independent from the histology of the primary tumor [111]. However, this statement did not match the findings in the larger INSPECT 2008–2019 cohort and even other small studies like the one by Kreuter et al., which have found a statistically significant poorer response of primary or secondary (stage III/IV) cSCC compared to other types of tumors (melanoma, lymphoma, or sarcoma) [100,112].

A retrospective study with 39 patients, 24 of them with SCC (either cSCC or oral cavity/oropharynx SCC) showed comparable complete responses of ECT regarding BCC or adenocarcinoma of the head and neck area, with mild toxicity [113]. There were better responses in small, primary, and naive cSCCs, with CR more likely being higher in the group treated with intratumoral bleomycin [113]. No responses were seen in patients treated with intralesional cisplatin in that study despite the fact that intratumoral cisplatin in the context of ECP has shown efficacy in a couple of small case series [106,113,114]. Cisplatin might be preferred in elderly patients and in those with renal disease, given that bleomycin has greater renal, skin, and lung toxicity. This safety profile should be considered when multiple treatments are needed [101,106]. Recently, two studies have shown a similar efficacy of ECT with reduced doses of intravenous bleomycin. A retrospective study of 13 SCCs and a nonrandomized prospective study included 7 SCCs in the experimental (reduced dose) group and 3 SCCs in the control (standard dose) group [115,116]. This approach could be especially useful in elderly patients with comorbidities [115,116].

ECT has also been used as a neoadjuvant setting in cSCCs located on the lower lip, in recurrent cSCCs on the scalp, in aesthetically compromised areas like the nose, and in cSCCs in particular conditions like recessive dystrophic epidermolysis bullosa and linear porokeratosis [117,118,119,120,121,122,123].

## 3. Emerging Treatments and Future Directions

### 3.1. Intralesional Immunotherapy

Immunotherapy (IT) has changed the therapeutic landscape of cSCC in recent years and it is currently the first line for advanced unresectable disease. Despite the fact systemic IT is generally well tolerated in patients with cSCC, even elderly people, a rationale for local delivery of the treatment to provide an even better safety profile and to expand the clinical scenarios under which IT can be used for cSCC was developed. Thus, intralesional/intratumoral approaches started to be evaluated. There are various intralesional drugs being tested in ongoing clinical trials (Table 5).

A phase 1 study with intralesional cemiplimab (anti-PD1, which blocks PD-1/PDL1 interaction, thus allowing T cells to kill tumor cells) is being carried out for cSCC and also BCC (NCT03889912) [124]. Vidutolimod is a CpG-A TLR9 (pattern recognition receptor expressed on endosomes internalized from the plasma membrane) agonist that is being tested in a phase 2 study. It is administered intratumorally in combination with intravenous cemiplimab in patients with advanced or metastatic cSCC (NCT04916002) [125].

CV8102 is a non-coding, non-capped RNA that activates the innate (via pattern recognition receptors TLR7/8, located in the intracellular endosomes, and RIG-I, located in the cytoplasm) and adaptive immunity that has been tested intratumorally in advanced cSCC (among other cancers) in a phase 1 study, in monotherapy and in combination with anti-PD1 (NCT03291002) [126]. Preliminary results showed it was well tolerated, with grade 1–2 fatigue, fever, chills, and headache being the most frequent adverse events, and demonstrated single-agent activity in three patients with melanoma and one with head and neck SCC [140].

A phase 1/1b study with E7766, a macrocycle-bridged stimulator of interferon genes (STING) agonist with antitumor activity, has been completed for patients with advanced, nonresectable, or recurrent solid tumors or lymphomas, and is pending results (NCT04144140) [127]. Daromun, a combination of two monoclonal antibody–cytokine fusion proteins, darleukin and fibromun (L19IL2 and L19TNF), is currently being tested in two phase 2 trials (NCT05329792, NCT04362722) that include non-metastatic cSCC patients and one of them also KA [128,129]. Giloralimab, an anti-CD40 monoclonal antibody, will be tested intratumorally in advanced solid tumors in one arm of a phase 1 study, with escalating doses (NCT02988960) [130]. TSN222 is a dual-action small molecule immune agonist with cytotoxic functions and is planned to be tested intratumorally in a phase 1/2 clinical trial that includes unresectable locally advanced or metastatic solid tumors (NCT05842785) [131,141]. SAR441000 is an mRNA mixture encoding IL-12 single chain, interferon alpha2b, GM-CSF, and IL-15sushi that has been tested intralesionally in a phase 1 study (NCT03871348) in two patients with advanced cSCC; only grade 1–2 adverse events were observed in preliminary results [132,142].

A novel drug, INT230-6, made from a cell permeation enhancer and classical chemotherapeutics cisplatin and vinblastine, has been tested in a phase 1/2 trial (NCT03058289) [133]. A total of 5 patients with advanced or metastatic SCCs were included: only 1/5 was alive at the end of follow-up and he had been treated with intralesional INT230-6 (>50% tumor burden) and pembrolizumab [143]. INT230-6 is well tolerated as a monotherapy or in combination with pembrolizumab [143].

### 3.2. Oncolytic Viruses

Talimogene laparhevec (T-VEC) is a herpes virus 1 oncolytic immunological agent approved for the treatment of metastatic melanoma by the FDA and EMA. There are some reports of its use in transplant patients with cSCC who are not candidates for immunotherapy due to the risk of graft rejection, two of them in liver transplant recipients with in-transit and metastatic cSCC and one in a renal transplant patient with multiple cSCCs, with good tolerance and clinical response [144,145,146]. T-VEC could be a promising treatment, especially for this subset of patients whose cSCC management is limited by the survival of the graft. It causes a local and systemic immunological response which not only would treat the injected lesions but could prevent the development of new cSCCs in this high-risk population subgroup [144,145,146]. Another phase 1 study of T-VEC in locally advanced cutaneous lymphomas and nonmelanoma skin cancers included one patient with locally advanced cSCC. Despite the fact the results per type of tumor were not detailed, overall the study showed a good response without unexpected adverse events [147]. A single-arm phase 2 study of T-VEC in low-risk cSCC has demonstrated an overall complete response with a favorable adverse event profile with mainly grade 1–2 transient fatigue, flu-like symptoms, and headaches [135].

Another type of oncolytic modified herpes simplex 1 virus, RP1, is included in two clinical trials: NCT05858229, a phase 1b trial studying treatment of resectable SCCs with intralesional RP1 as a neo-adjuvant, and NCT04050436, a phase 2 trial comparing intralesional RP1 vs. intralesional RP1 with cemiplimab in patients with locally advanced or metastatic disease [134,148]. Also, a phase 1b/2 clinical trial with gebasaxturev, an oncolytic coxsackievirus A21, is ongoing (NCT04521621) [136]. Patients with locally advanced or metastatic SCC will receive eight cycles of intratumoral gebasaxturev in combination with pembrolizumab [136]. Finally, TBio-6517, an oncolytic Vaccinia Virus, is currently immersed in a phase 1/2a trial that includes patients with locally advanced or metastatic SCC that will receive pembrolizumab and intratumoral Tbio-6517 (NCT04301011) [137]. Ongoing clinical trials are summarized in Table 5.

Interestingly, there are two recent reports on the use of intralesional and systemic Human Papillomavirus vaccine in elderly women with cSCC (one of them with multiple cSCCs on the leg), with complete clearance of the tumors [149,150]. It could become a safe and widely available treatment option in inoperable tumors or patients who are not candidates for surgery.

### 3.3. Cancer Vaccines

IFx-Hu2.0, an intralesional personalized cancer vaccine that contains emm55 (an immunogenic bacterial protein) was designed to overcome primary resistance to checkpoint inhibitors by making cancer cells look like they were bacteria, then triggering the innate immune response [151]. It has been tested in a phase 1 clinical trial to evaluate its safety in melanoma, SCC, and basal cell carcinoma, pending results (NCT04925713) [138]. Another phase 1 clinical trial for patients with locally advanced nonmelanoma skin cancer which had progressed despite prior therapy with immune checkpoint inhibitors (ICI) is still recruiting, but preliminary data, which included four patients with advanced SCC, have shown that this approach could make their tumors respond to ICI even with the prior failure of this drugs with a good safety profile (NCT04160065) [139,152]. Ongoing clinical trials are summarized in Table 5.

## 4. Conclusions

Despite surgery remaining the preferable treatment for cSCC and KAs, intralesional treatments for cSCC and KA pose an alternative therapeutic option especially for patients who are not willing to or cannot undergo surgery due to fragility or comorbidities (Figure 1 and Figure 2). Although MTX has been the most frequently used drug in the literature, many other options are available and their use will depend on the patient’s profile, their preferences, the physician’s experience, and the available health resources. Many new promising molecules are under research.

## Figures and Tables

**Figure 1 cancers-16-00158-f001:**
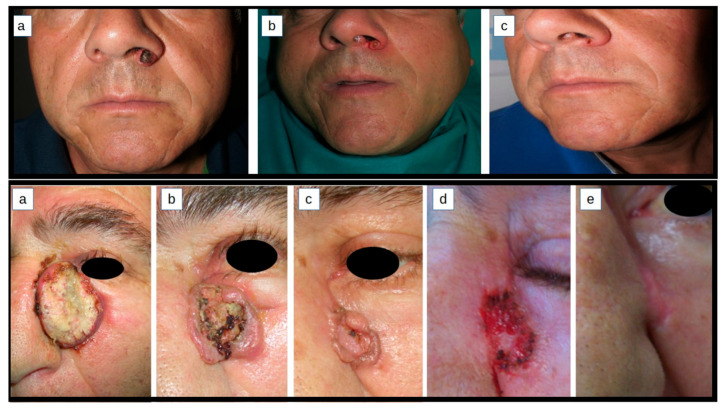
First row of images shows a case of KA located on the left ala of the nose before, during, and after treatment with ilMTX (3 injections of 10 mg of MTX, 1 week apart). Second row of images shows a case of KA located on the left nasogenian fold before, during and after treatment with ilMTX (4 injections of 10 mg of MTX, 1 week apart). Both patients achieved complete response.

**Figure 2 cancers-16-00158-f002:**
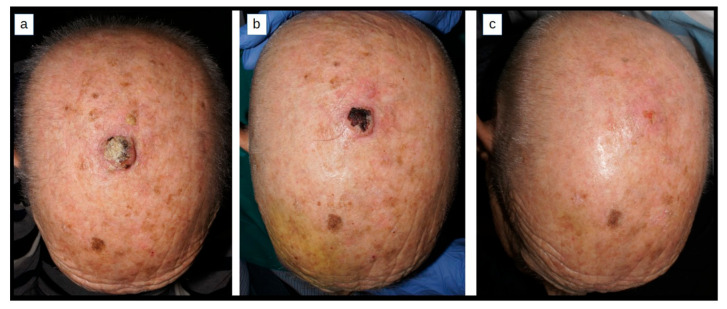
This 86-year-old male with a cSCC on the scalp refused surgery; these pictures show before, during, and after treatment with ilMTX (2 injections of 10 mg of MTX, 1 week apart). He achieved complete response and no recurrence was observed during the following 3 years.

**Table 1 cancers-16-00158-t001:** Intralesional methotrexate for treatment of keratoacanthoma (modified from Kiss et al. and Searle et al.) [7,8].

Author, Year	*n*	Location	MTX Administration	Outcome	Adverse Events Related to ilMTX
Melton et al., 1991 [12]	9	Face/scalp (7) > trunk (1) and hand (1)	1–2 injections; cumulative dose: 5.0–37.5 mg of MTX	100% resolution	Discomfort with injection
Hurst et al., 1995 [13]	2	Nose	1–3 injections; cumulative dose: 25–75 mg of MTX	100% resolution	None reported
Hong et al., 1997 [14]	1	Lower lip	3 injections; cumulative dose: 75 mg of MTX	100% resolution	None reported
Cuesta-Romero et al., 1998 [15]	6	Nose (5) > Cheek (1)	1–4 injections; cumulative dose: 12.5–62.5 mg of MTX	100% resolution	None reported
Spieth et al., 2000 [16]	1	Lower lip	5 injections; cumulative dose: 25 mg of MTX	100% resolution	Moderate pain with injection
Remling et al., 2000 [17]	1	Nose	2 injections; cumulative dose: 21.25 mg of MTX	100% resolution	None reported
Kim et al., 2001 [18]	1	Nose	2 injections; cumulative dose not reported	100% resolution	None reported
You et al., 2022 [19]	2	Thigh and cheek	2–6 injections; cumulative dose: 15–45 mg of MTX	100% resolution	None reported
De Visscher et al., 2002 [20]	1	Lower lip	2 injections; cumulative dose: 50 mg of MTX	100% resolution	None reported
Cohen et al., 2005 [21]	1	Nose	3 injections; cumulative dose: 19.5 mg of MTX	100% resolution	None reported
Shin et al., 2006 [22]	2	Lower lip and forehead	6 injections; cumulative dose: 21–120 mg of MTX	100% resolution	None reported
Yuge et al., 2006 [23]	1	Leg (KA marginatum centrifugum)	3 injections; cumulative dose: 37.5 mg of MTX	Failure; resolution with topical 5-FU.	None reported
Annest et al., 2007 [24]	18	Face/scalp (10) > leg (4) and hand (4)	1–3 injections; cumulative dose: 2.0–87.5 mg of MTX	83% resolution	None reported
Basoglu et al., 2008 [25]	1	Upper lip	10 injections; cumulative dose: 85 mg	Recurrence after 3 months (neurotropic KA)	None reported
		After CO_2_ laser			
Martorell-Calatayud et al., 2011 [9]	10	Face (7) > dorsum of hand (3)	1 neoadjuvant injection prior to surgery; 0.3–0.5 mL of 25 mg/mL of MTX	Tumor size reduction 50–80%, 100% surgical direct wound closure	None reported
Patel et al., 2011 [26]	9	Face (5) > leg (3) > arm (1)	1–4 injections; 12.5–25.0 mg of MTX	88.9% resolution	Transient injection pain (*n* = 1)
Aubut et al., 2012 [27]	46	Head (35) > other	1–4 injections; cumulative dose: 2–25 mg of MTX	74% resolution	None reported
				11/12 non-responders SCC histology on surgical excision	
Yoo et al., 2014 [28]	5	Nose (2) > cheek, chin, lower lip	2–7 injections; cumulative dose: 25–90 mg of MTX	80% resolution and 20% excision after reduction	None reported
Panther et al., 2015 [29]	1	5 lesions on the face	1 injection; 0.2–0.3 mL of 12.5 mg/mL of MTX per lesion	100% resolution	None reported
		After CO_2_ laser	with topical imiquimod		
Veerula et al., 2016 [30]	1	Leg, over scar of SCC (KA marginatum centrifugum-like, isotopic and isomorphic response)	3 injections; cumulative dose: 37.5 mg of MTX	100% resolution	None reported
Rambhia et al., 2017 [11]	1	Multiple KAs on the legs, buttocks, elbow, and hand	2 injections; 0.3–2.0 mL of 12.5 mg/mL of MTX	Shrinkage of lesions	Pain at site of injection
			with acitretin	Later treated with topical 5-FU and tazarotene	
Rossi et al., 2017 [31]	14	Head and neck	1–3 injections; 12.5–25.0 mg of MTX	71.4% resolution	None reported
Barros et al., 2017 [32]	1	KA-like lesions of incontinentia pigmenti on the leg	3 injections; 12.5–43.75 mg of MTX	100% resolution	Injection discomfort
Della Valle et al., 2018 [33]	1	Dorsum of hand	2 injections; cumulative dose: 40 mg of MTX	100% resolution	None reported
Moss et al., 2019 [10]	54	73 tumors	1–4 injections; cumulative dose: 1.3–31.3 mg of MTX	88% resolution	None reported
		Leg (60) > arm (9) > trunk (2) > head and neck (2)			
Scalvenzi et al., 2019 [34]	11	13 lesions	4–8 injections; 20 mg of MTX (≤2 cm, *n* = 6) or 25 mg of MTX (>2 cm, *n* = 5)	100% resolution	None reported
		Face (6) > trunk (3) > ears (2) and hands (2)			
Saporito et al., 2019 [35]	3	KA and well-differentiated SCC	2–3 injections of 25 mg/mL of MTX	100% resolution	None reported
Doerfler et al., 2019 [36]	1	Nose	3 injections; cumulative dose: 75 mg of MTX	100% resolution	None reported
Smith et al., 2020 [37]	29	69 lesions	Mean: 2 injections; mean cumulative dose: 39 mg of MTX	95.7% resolution	None reported
		Leg (62) > arm (7)			
Gualdi et al., 2020 [38]	12	Not reported	4–6 injections (different protocols, not separated by type of tumor); cumulative mean dose: 133.29 mg of MTX	92% resolution	Yes; anemia the most frequent (5/35)

SCC stands for squamous cell carcinoma, KA stands for keratoacanthoma and MTX stands for methotrexate.

**Table 2 cancers-16-00158-t002:** Intralesional methotrexate for treatment of invasive squamous cell carcinoma (modified from Searle et al.) [8].

Author, Year	*n*	Location	MTX Administration	Outcome	Adverse Events Related to ilMTX
Plascencia-Gómez et al., 2014 [44]	1	Lower lip	3 injections of MTX, 1 week apart before surgery; cumulative dose: 75 mg of MTX	80% tumor size reduction	None reported
Moye et al., 2014 [45]	1	Multiple SCCs (face, thigh, back, and calf) Patient with melanoma receiving vemurafenib	4 injections of MTX, 3–4 weeks apart; approximately 1.9–12.5 mg of MTX per injection	All tumors decreased in size	Tolerable pain with injection
Salido-Vallejo et al., 2016 [39]	43	Cheek (12) > scalp (8) > temple (6) > lower lip (5), arm (5) > nose (3), ear (3) > leg (1)	1 injection of 25 mg/mL MTX before surgery; mean: 0.74 mL (0.1–1.3 mL)	Reduction in tumor size (mean: 42.6%) compared to surgery alone; greater reduction in lower lip tumors	Discomfort during MTX infiltration (60.47%)
Bergón-Sendín et al., 2018 [40]	10	Lower lip	2 injections of 20 mg MTX, 1 week apart before scheduled surgery; cumulative dose: of MTX 40 mg	100% response Mean decrease in diameter: 68.2%	None reported
Bergón-Sendín et al., 2019 [43]	40	Face (50%) > extremities (22.5%) > scalp (17.5%) > trunk (2.5%)	2 injections of 50 mg/mL MTX, 1 week apart before surgery; mean cumulative dose: 37.6 mg	Clinical and sonographic response in 92.5%	None reported
Bergón-Sendín et al., 2020 [41]	84	Not reported	2 injections of MTX before scheduled surgery, 1 week apart; mean cumulative dose: 36.9 mg of MTX	100% tumor size reduction; 46/84 complete clinical and histopathological response	None reported
Gualdi et al., 2020 [38]	21	Not reported	4–6 injections (different protocols, not separated by type of tumor); mean cumulative dose: 133.29 mg of MTX	47.6% resolution	Yes; anemia the most frequent (5/35)
Bergón-Sendín et al., 2021 [42]	100	Face (50) > scalp (18) > limbs (16) > lip/ear (14) > trunk (2)	2 injections of MTX, 1 week apart before scheduled surgery; mean cumulative dose: 37.72 mg of MTX	93% tumor size reduction; less complex reconstructions than surgery-only group	None reported
Vega-González et al., 2022 [46]	1	Lower lip	3 injections of MTX monthly; cumulative dose: 75 mg of MTX	100% resolution	None reported

SCC stands for squamous cell carcinoma, KA stands for keratoacanthoma and MTX stands for methotrexate.

**Table 3 cancers-16-00158-t003:** Intralesional 5-fluorouracil for treatment of keratoacanthoma (modified from Kiss et al., Metterle et al. and Maghfour et al. [7,50,51]).

Author, Year	*n*	Location	5-FU Administration	Outcome	Adverse Events Related to il5-FU
Klein et al., 1962 [64]	2	Ear, nose	0.1 mL of 5% il5FU, twice daily or every other day; 7–34 injections	100% resolution	Not reported
Odom et al., 1978 [65]	14	26 KAs on face and upper extremities	0.2–0.5 mL of 5% il5-FU, once a week for 2–8 weeks (mean: 2.8 injections)	96.2% resolution	Mild pain during injection, slight irritation, and necrotic involution of KAs
Goette et al., 1980 [52]	30	41 KAs (arms > face > shins)	0.5–1.0 mL of 5% il5FU, 1 week apart; up to 5 injections (mean: 3 injections)	97.5% resolution	Mild pain during injection, slight irritation, and necrotic involution of KAs
Kurtis et al., 1980 [66]	3	Upper lip, medial epicanthus, neck	Cumulative dose of 3.55–11.7 mL of 5% il5FU in 6–12 injections	100% resolution	Necrotic involution of KA
Eubanks et al., 1982 [67]	1	Multiple KAs on the arms, also over scar	0.1–0.2 mL of 5% il5FU per lesion, 1 week apart; 5–9 injections	100% resolution	None reported
Parker et al., 1986 [68]	5	Face	1–3 mL of 5% il5FU, 1–4 weeks apart; 2–6 injections	100% resolution	None reported
Bergin et al., 1986 [69]	1	Eyelid	0.5 and 0.25 mL of 5% il5FU, 1 week apart	100% resolution	Minimal pain with injection and edema
Singal et al., 1997 [70]	1	Multiple lesions on buttocks and legs (probably Ferguson-Smith type)	0.2–0.3 mL of 5% il5-FU, 1 week apart for 3 weeks (only larger lesions)	Nearly complete resolution	None reported
Leonard et al., 2006 [71]	1	Nose	8 injections of 5-FU (50 mg/mL), 1–2 weeks apart for 14 weeks	100% resolution	None reported
Hadley et al., 2009 [63]	3	Multiple KAs over scar	1 to 2 mL of 5% il5-FU, 1 week apart for 8 weeks with acitretin in 1 case	2/3 resolution 1/3 discontinued	1 case of shortness of breath during injection
LaPresto et al., 2013 [62]	1	13 lesions (cheek, shoulder, chest, back, and leg) Patient with melanoma receiving vemurafenib	1 injection of 5% il5-FU, 2.5 mL (mean: 0.2 mL per lesion) with acitretin	Nearly complete resolution and significant reduction in size	None reported
Que et al., 2018 [72]	30	136 lesions (eruptive KA) Legs (80%) > arms (10%) and arms/legs (10%)	Median of 2 injections (1–8) of 0.5 mL (0.1–1.0) of 5% il5-FU per lesion every 3 weeks (2.0–8.5)	Focal koebnerizing eruptive KA: 91% complete resolution; diffuse KA type: 53% complete resolution	Mild; cutaneous dyspigmentation and shallow erosions
Dominiak et al., 2016 [73]	1	Leg (type KA centrifugum marginatum)	2 injections of 0.6 mL of 5% il-5FU, 3 weeks apart	100% resolution	Discomfort with injection
Hemperly et al., 2020 [74]	1	2 lesions on arm (field cancerization, previous radiation)	4–5 injections of 5% il5-FU, 1 week apart; cumulative doses: 22.5 mg and 32.5 mg of 5FU, respectively	100% resolution	Tolerable pain during injection
Seger et al., 2020 [75]	1	Arm (eruptive KA over scar)	4 injections of 5-FU, every 3–4 weeks; total of 5 mL of 5-FU	100% resolution	None reported
Hamad et al., 2021 [59]	46	Not separated by tumor type; >legs	Mean of 1.6 injections of 0.1–1.0 mL of 5% 5-FU (max. 1 mL per 3 lesions), 2–8 weeks apart Prior debulking	97.8% resolution	Temporary alopecia, nausea, fatigue, and dyspigmentation
Ahmed et al., 2022 [76]	1	11 lesions (eruptive KA) on the legs after COVID-19 vaccine	1.5 mL of 5% 5-FU per lesion, once every week for 3 weeks	100% resolution	None reported
Yumeen et al., 2023 [61]	1	2 lesions (eruptive KA) Patient with IgG immunodeficiency	1 injection; dose not reported with acitretin and topical 5-FU/imiquimod	100% resolution	None reported
Marka et al., 2023 [53]	7	40 KAs, >legs	2–13 injections of dilute 5-FU (10.0–16.7 mg/mL), 1 week to one month apart; mean dose per visit: 3.3–32.5 mg of 5-FU	97.5% resolution	None reported

KA stands for keratoacanthoma, 5FU stands for 5-fluorouracile and il stands for intralesional.

**Table 4 cancers-16-00158-t004:** Intralesional 5-fluorouracil for treatment of invasive SCC (modified from Metterle et al. and Maghfour et al.) [50,51].

Author, Year	*n*	Location	5-FU Administration	Outcome	Adverse Events Related to il5-FU
Kraus et al., 1998 [77]	23	Head and neck (10) > upper extremities (9) > trunk (4)	4–6 injections of 3% 5-FU gel, 1 week apart; median cumulative dose: 3.7 mL (0.6–6)	96% complete resolution	None reported
Morse et al., 2003 [78]	1	Face	8 injections of 5-FU, 1 week apart; weekly dose: 0.8–2.4 ml	100% resolution	None reported
Khandpur et al., 2003 [60]	1	3 SCCs on thumb, toe, and ankle	Injection of 1 mL (per lesion) of 5FU 0.05 mg/mL, 4 days per week for 4 weeks	100% resolution	None reported
		Patient with arsenical keratosis	with acitretin		
Reisinger et al., 2011 [79]	1	Thumb	6 injections of 5% 5-FU, 1 week apart; cumulative dose: 600 mg of 5-FU	100% resolution	None reported
Mackey et al., 2018 [80]	1	5 SCCs and 1 atypical squamous proliferation on the legs	6 injections of 0.6 mL of 5% 5-FU per lesion, 1 week apart; cumulative dose: 180 mg per lesion	100% resolution	None reported
Manalo et al., 2019 [58]	2	Multiple lesions on the legs	4–12 injections of 0.1–0.5 mg 5FU, 1 week apart	100% resolution	Ulcer at injection sites; infected ulcer
			with chemowraps of 5% 5-FU and acitretin		
Dando et al., 2020 [57]	3	7 lesions on the legs (6) and arm (1)	1–2 injections of 0.3–1.0 mL 5% 5-FU	85.7% resolution	Local pain (1) and mild pruritus (1)
Hamad et al., 2021 [59]	230	Not classified by tumor type; >legs	Mean of 1.6 injections of 0.1–1.0 mL of 5% 5-FU (max. 1 mL per 3 lesions), 2–8 weeks apart	83% resolution in invasive SCC; higher resolution in well-differentiated SCC (90%)	Temporary alopecia, nausea, fatigue, and dyspigmentation
			Prior debulking		
Maxfield et al., 2021 [54]	148	172 lesions (7/172 KA subtype), >lower extremity (37%)	Mean of 1.25 injections of 0.2–2.0 mL 5% 5-FU per lesion, 4 weeks apart	92% resolution	*n* = 5, mostly local site reactions; 1 patient with headache, dizziness, and nausea
Marka et al., 2023 [53]	4	10 lesions, >legs	1–6 injections of dilute 5-FU (10.0–16.7 mg/mL), 1 week to 1 month apart; mean dose per visit: 8.3–13.4 mg of 5-FU	90% resolution	None reported
Luu et al., 2023 [81]	15	23 lesions, >legs	Average of 4 injections (1–35) of 75 mg (50–150 mg) of dilute 5% 5-FU	87% resolution and 8.7% size reduction	Ulceration (6) and infection (4)

KA stands for keratoacanthoma, 5FU stands for 5-fluorouracile and il stands for intralesional.

**Table 5 cancers-16-00158-t005:** Ongoing trials in emerging treatments, pending results.

Drug		Treatment	Mechanism of Action	Condition	Current Status	Phase	NCT Code
Immunotherapy	Cemiplimab	Alone	Anti-PD1	cSCC	Recruiting	Phase 1	NCT03889912 [124]
		Cemiplimab with vidutolimod	Anti-PD1 with CpG-A TLR9	Advanced or mcSCC	Recruiting	Phase 2	NCT04916002 [125]
	CV8102	Alone or with anti-PD1	TLR7/8 and RIG-I	Advanced cSCC	Active, not recruiting	Phase 1	NCT03291002 [126]
	E7766	Alone	STING agonist	Advanced solid tumors	Completed	Phase 1/1b	NCT04144140 [127]
	Daromun	Alone	L19IL2/L19TNF	Non-metastatic cSCC and KA	Recruiting	Phase 2	NCT05329792 [128]
							NCT04362722 [129]
	Giloralimab	Alone or with budigalimab	Anti-CD40	Advanced solid tumors	Active, not recruiting	Phase 1	NCT02988960 [130]
	TSN222	Alone	Bifunctional small molecule	Unresectable locally advanced or metastatic solid tumors	Not yet recruiting	Phase 1/2	NCT05842785 [131]
	SAR441000	Alone or with cemiplimab	mRNA mixture encoding IL-12 single chain, interferon alpha-2b, GM-CSF, and IL-15sushi	Advanced solid tumors	Active, not recruiting	Phase 1	NCT03871348 [132]
	INT230-6	Alone or with anti-PD1	Cell permeation enhancer and cisplatin and vinblastin	Advanced or metastatic cSCC	Completed	Phase 1/2	NCT03058289 [133]
Oncolytic viruses	RP1	Alone (neoadjuvant)	Oncolytic modified herpes simplex 1	Resectable cSCC	Not yet recruiting	Phase 1b	NCT05858229 [134]
		Alone or with cemiplimab	Oncolytic modified herpes simplex 1	lacSCC and mcSCC	Active, not recruiting	Phase 2	NCT04050436 [135]
	Gebasaxturev	With pembrolizumab	Oncolytic coxsackievirus A21	Advanced or metastatic SCC	Completed	Phase 1b/2	NCT04521621 [136]
	TBio-6517	With pembrolizumab	Oncolytic vaccinia virus	Locally advanced or metastatic SCC	Active, not recruiting	Phase 1/2a	NCT04301011 [137]
Cancer vaccines	IFx-Hu2.0	Alone	Bacterial protein emm55	cSCC	Completed	Phase 1	NCT04925713 [138]
		Alone		Advanced nonmelanoma skin cancer	Recruiting	Phase 1	NCT04160065 [139]

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
