# Peer review of "Intralesional Treatments for Invasive Cutaneous Squamous Cell Carcinoma"

_cancers, 2023, doi:10.3390/cancers16010158_

Round 1
Reviewer 1 Report
Comments and Suggestions for Authors
1#. Please define invasive cutaneous squamous cell carcinoma in background introduction. Although the title is about Intralesional treatments for invasive cutaneous squamous cell carcinoma, the entire paper is focused on cutaneous squamous cell carcinoma (cSCC) and keratoacanthoma (KA). Does all cutaneous squamous cell carcinoma belong to invasive cutaneous squamous cell carcinoma?
2#. In the past, keratoacanthoma was often considered a low-grade variant of squamous cell carcinoma. However, it's important to note that the distinction between keratoacanthoma and well-differentiated squamous cell carcinoma can be somewhat arbitrary and has been the subject of debate in the medical community. Some experts classify keratoacanthoma as a separate entity, while others consider it as a variant of well-differentiated squamous cell carcinoma. Is keratoacanthoma a kind of invasive cutaneous squamous cell carcinoma? If not, the title of invasive cutaneous squamous cell carcinoma is inappropriate.
3#. Please define Intralesional treatments in background introduction and introduce its advantages, such as Targeted Treatment: It allows for precise delivery of medication directly to the affected area, minimizing systemic exposure. Reduced Side Effects: Because the medication is localized, it may lead to fewer systemic side effects compared to oral or intravenous administration. Avoidance of Surgery: In some cases, intralesional therapy can be an alternative to surgical intervention.
4#. According to the first literature citation, it’s said “After basal cell carcinoma, cSCC is the second most common skin tumor and constitutes 20% of all cases of non-melanocytic skin cancer (NMSC).” In the abstract and introduction of this review, cutaneous squamous cell carcinoma is described as “the second most frequent cancer in humans”, which is really not rigorous. Please correct it.
5#. Please add literature citation and offer specific information for background knowledge, such as “cSCC tends to exhibit benign clinical behavior, but has the potential to progress locally, metastasize, and cause death in a subset of patients. cSCC is especially common in the elderly and as life expectancy increases it will probably continue to represent a major health concern in the population in the future.” Please clear the population of dead patients and age range of patients with cSCC, instead of the word subset or elderly.
6#. In the line of 151, Kas should be corrected to KAs.
7#. In the line of 217, there is something wrong with the 3, 57 citations.
8#. In lines of 291-293, fonts are inconsistent.
9#. In the line of 334, 3.4. should be corrected to 3.3.
10#. There is no outcome information in Table 5. Please add or explain it.
Comments on the Quality of English Language
The article is well organized and its presentation is good and easy to understand.
Author Response
Reviewer #1
1#. Please define invasive cutaneous squamous cell carcinoma in background introduction. Although the title is about Intralesional treatments for invasive cutaneous squamous cell carcinoma, the entire paper is focused on cutaneous squamous cell carcinoma (cSCC) and keratoacanthoma (KA). Does all cutaneous squamous cell carcinoma belong to invasive cutaneous squamous cell carcinoma?
ANSWER: Many thanks for your suggestion. We have clarified the concept in the introduction. In this review, we focus on invasive (not in situ) cutaneous squamous cell carcinoma (CSCC) and on keratoacanthoma. There is some controversy on the nosologic position of keratoacanthoma (KA) and some authors have suggested it is an entity different from CSCC, while others claim KA is indeed a variant of CSCC. Notwithstanding that, treatment strategies are similar and, in most papers, both CSCC and KA are considered together. For that reason, we focus in both these entities. In situ CSCC is excluded as it will be covered elsewhere in this Special Issue.
Page 1, Lines 26-30 “There is some controversy on the nosologic position of keratoacanthoma and some authors have suggested it is an entity different from cSCC, while others claim KA is indeed a variant of cSCC. Notwithstanding that, treatment strategies are similar and, in most papers of intralesional treatments, both cSCC and KA are considered together7. For that reason, we focus in both these entities.”
2#. In the past, keratoacanthoma was often considered a low-grade variant of squamous cell carcinoma. However, it's important to note that the distinction between keratoacanthoma and well-differentiated squamous cell carcinoma can be somewhat arbitrary and has been the subject of debate in the medical community. Some experts classify keratoacanthoma as a separate entity, while others consider it as a variant of well-differentiated squamous cell carcinoma. Is keratoacanthoma a kind of invasive cutaneous squamous cell carcinoma? If not, the title of invasive cutaneous squamous cell carcinoma is inappropriate.
ANSWER: Thank you for the appreciation. As you have stated, the distinction between invasive cutaneous squamous cell carcinoma and keratoacanthoma can be difficult. We have added a sentence in the paper trying to explain our decision on including both CSCC and KA.
Page 1, Lines 26-30 “There is some controversy on the nosologic position of keratoacanthoma and some authors have suggested it is an entity different from cSCC, while others claim KA is indeed a variant of cSCC. Notwithstanding that, treatment strategies are similar and, in most papers of intralesional treatments, both cSCC and KA are considered together7. For that reason, we focus in both these entities.”
3#. Please define Intralesional treatments in background introduction and introduce its advantages, such as Targeted Treatment: It allows for precise delivery of medication directly to the affected area, minimizing systemic exposure. Reduced Side Effects: Because the medication is localized, it may lead to fewer systemic side effects compared to oral or intravenous administration. Avoidance of Surgery: In some cases, intralesional therapy can be an alternative to surgical intervention.
ANSWER: We have added some sentences in the introduction regarding the advantages of intralesional therapies in certain contexts.
Page 1, Line 19-23. “Intralesional treatments are administered directly inside the tumor, thus providing a selective drug delivery which prevents most systemic side effects6. This approach may be especially useful in patients with comorbidities and in those cases in which surgery is likely to result in either functional or cosmetic impairment6.”
4#. According to the first literature citation, it’s said “After basal cell carcinoma, cSCC is the second most common skin tumor and constitutes 20% of all cases of non-melanocytic skin cancer (NMSC).” In the abstract and introduction of this review, cutaneous squamous cell carcinoma is described as “the second most frequent cancer in humans”, which is really not rigorous. Please correct it.
ANSWER: Thank you for your comment. It seems that part on the introduction was not clear. It has now been completely rewritten.
Page 1, lines 3-15. “Cutaneous squamous cell carcinoma (cSCC) derives from the malignant proliferation of keratinocytes1. cSCC may arise de novo or from potential precancerous lesions such as actinic keratosis, actinic cheilitis or chronic inflammatory diseases2. While it usually exhibits benign clinical behavior, cSCC has the potential to progress locally, metastasize, and cause death in a subset of patients1. Ten-year survival after surgery exceeds 90% for cSCC but drops dramatically when metastases occur3. The frequency of lymph node metastases is around 4%, and mortality rates are nearly 2%. Given its high frequency, cSCC has a significant impact on overall mortality4. It is responsible for the majority of deaths from skin cancer in people older than 85 years and in some areas of the US, it has a mortality comparable to that of renal, oropharyngeal, and melanoma carcinomas5. cSCC is especially common in the head and neck area of males older than 65 years due to chronic ultraviolet light exposure and as life expectancy increases, it will probably continue to represent a major health concern in the future2.”
5#. Please add literature citation and offer specific information for background knowledge, such as “cSCC tends to exhibit benign clinical behavior, but has the potential to progress locally, metastasize, and cause death in a subset of patients. cSCC is especially common in the elderly and as life expectancy increases it will probably continue to represent a major health concern in the population in the future.” Please clear the population of dead patients and age range of patients with cSCC, instead of the word subset or elderly
ANSWER: We have clarified this topic in the text.
Page 1, lines 5-15 “While it usually exhibits benign clinical behavior, cSCC has the potential to progress locally, metastasize, and cause death in a subset of patients1. Ten-year survival after surgery exceeds 90% for cSCC but drops dramatically when metastases occur3. The frequency of lymph node metastases is around 4%, and mortality rates are nearly 2%. Given its high frequency, cSCC has a significant impact on overall mortality4. It is responsible for the majority of deaths from skin cancer in people older than 85 years and in some areas of the US, it has a mortality comparable to that of renal, oropharyngeal, and melanoma carcinomas5. cSCC is especially common in the head and neck area of males older than 65 years due to chronic ultraviolet light exposure and as life expectancy increases, it will probably continue to represent a major health concern in the future2.”
6#. In the line of 151, Kas should be corrected to KAs.
ANSWER: Thank you. We have checked that all abbreviations standing for keratoacanthoma appear as KAs.
7#. In the line of 217, there is something wrong with the 3, 57 citations.
ANSWER: Thank you for the appreciation. The reference is number 57. The 3 refers to cm3. It has now been corrected.
8#. In lines of 291-293, fonts are inconsistent.
ANSWER: Thank you for the appreciation. We have now checked that fonts are consistent both concerning style and size.
9#. In the line of 334, 3.4. should be corrected to 3.3.
ANSWER: Thank you for the appreciation. We have now checked that sections are consistent throughout the manuscript. Should you find any additional error, please let us know.
10#. There is no outcome information in Table 5. Please add or explain it.
ANSWER. Table 5 refers to ongoing trials for which results are still lacking. Thus, we are not able to provide information on the outcomes. We have clarified this in the legend.
Reviewer 2 Report
Comments and Suggestions for Authors
The subject is of interest, however, this review requires revisions prior considerations. Specifics are below.
Proper introduction of to the problem of cutaneous SCC is missing.
The authors mention immunosuppression but ignore the main etiological factor UVR, specifically UVB. This has to be improved as relates to the etiology. Also, readers would appreciate information on diverse effects of UVR in the skin (Endocrinology 159(5), 1992-2007, 2018).
Some overview of epidermal carcinogenesis is required with description of progression of SCC.
Precancerous states such as solar keratoses should be discussed.
The main mode of action is biopsy and surgical re-excision if necessary. Therefore, better line of argumentation for other types of treatment is expected.
Photodynamic therapy section is rather short.
Different pharmacological and biological methods of therapy could be costly (except traditional ones) and have multiple site effects. Therefore I am surprised that potential use of vitamin D derivatives is not mentioned.
Many of them have photoprotective (Cell Biochem Biophys 78(2):165-180, 2020) and anticarcinogenic in the skin (International Journal of Oncology 2022;61(2). doi: ARTN 9610.3892/ijo.2022.5386; Adv Exp Med Biol. 1268, 257-283, 2020) effects.
Comments on the Quality of English LanguageMinor editing of English language required
Author Response
Reviewer #2
-Proper introduction of to the problem of cutaneous SCC is missing.
ANSWER: Thank you for your comment. It seems that part on the introduction was not clear. It has now been completely rewritten. Concerning the background on CSCC, now it says.
Page 1- Lines 1-15: “Cutaneous squamous cell carcinoma (cSCC) derives from the malignant proliferation of keratinocytes1. cSCC may arise de novo or from potential precancerous lesions such as actinic keratosis, actinic cheilitis or chronic inflammatory diseases2. While it usually exhibits benign clinical behavior, cSCC has the potential to progress locally, metastasize, and cause death in a subset of patients1. Ten-year survival after surgery exceeds 90% for cSCC but drops dramatically when metastases occur3. The frequency of lymph node metastases is around 4%, and mortality rates are nearly 2%. Given its high frequency, cSCC has a significant impact on overall mortality4. It is responsible for the majority of deaths from skin cancer in people older than 85 years and in some areas of the US, it has a mortality comparable to that of renal, oropharyngeal, and melanoma carcinomas5. cSCC is especially common in the head and neck area of males older than 65 years due to chronic ultraviolet light exposure and as life expectancy increases, it will probably continue to represent a major health concern in the future2.”
-The authors mention immunosuppression but ignore the main etiological factor UVR, specifically UVB. This has to be improved as relates to the etiology. Also, readers would appreciate information on diverse effects of UVR in the skin (Endocrinology 159(5), 1992-2007, 2018).
ANSWER. Many thanks for your comment. This paper has not focused on risk factors for CSCC development, of which UVR is the most relevant. Immunosuppression was instead referred as a patient feature which might make him/her more fragile and amenable for intralesional treatment. Also, we don’t think this reference is sound with this paper. If the editor considers it relevant, we might include accordingly.
-Some overview of epidermal carcinogenesis is required with description of progression of SCC. Precancerous states such as solar keratoses should be discussed.
ANSWER: We have modified our introduction paragraph to include some precancerous entities that can transform into cutaneous squamous cell carcinoma. Molecular pathways of epidermal carcinogenesis are beyond our scope for space limitations. However, they are reasonably explained in reference 1.
Page 1, lines 4-5 “ cSCC may arise de novo or from potential precancerous lesions such as actinic keratosis, actinic cheilitis or chronic inflammatory diseases2”
-The main mode of action is biopsy and surgical re-excision if necessary. Therefore, better line of argumentation for other types of treatment is expected.
ANSWER: Many thanks for your appreciation. We have added some sentences in the introduction regarding the advantages of intralesional therapies in certain contexts.
Page 1, Line 19-23. “Intralesional treatments are administered directly inside the tumor, thus providing a selective drug delivery which prevents most systemic side effects6. This approach may be especially useful in patients with comorbidities and in those cases in which surgery is likely to result in either functional or cosmetic impairment6.”
-Photodynamic therapy section is rather short.
ANSWER: We agree with your comment. However, as we are only referring to intralesional photodynamic therapy, the available information is scarce. We have specified in the title “Intralesional photodynamic therapy” to make it clear that we are only referring to this modality of photodynamic therapy.
-Different pharmacological and biological methods of therapy could be costly (except traditional ones) and have multiple site effects. Therefore I am surprised that potential use of vitamin D derivatives is not mentioned. Many of them have photoprotective (Cell Biochem Biophys 78(2):165-180, 2020) and anticarcinogenic in the skin (International Journal of Oncology 2022;61(2). doi: ARTN 9610.3892/ijo.2022.5386; Adv Exp Med Biol. 1268, 257-283, 2020) effects.
ANSWER. Many thanks for your suggestions. We have read these papers with great interest. There is some evidence on the use of calcipotriol in combination with 5FU, despite topically applied and only for actinic keratoses. However, we are concerned that vit D is an area of extensive research in this field. However, we were unable to find information on intralesional either clinical trials or treatments with vitamin D and its derivates in CSCC and for that reason we considered not totally justified to cite any of this excellent references.
Round 2
Reviewer 2 Report
Comments and Suggestions for Authors
The authors partially addressed the critique. Please fully address the critique.
Comments on the Quality of English LanguageMinor editing of English language required
Author Response
Dear Editors of Cancers journal,
Many thanks for your editorial email with the reviewers’ comments and suggestions to our manuscript entitled “Intralesional treatments for invasive cutaneous squamous cell carcinoma”. We have carefully read the reviewers’ comments and tried to answer accordingly. We thank the reviewer for his/her suggestions, which we believe have helped us to improve our manuscript.
We are not able to find from the reviewer’s comments nothing concrete points on the manuscript to be addressed, beyond minor English editing, which we have tried to accomplish. Please, see below our response to reviewer’s comments.
We hope Cancers journal considers this as an interesting paper to be published.
We are looking forward to hearing from you soon.
Javier Cañueto, MD, PhD
Reviewer comments
COMMENT: The authors partially addressed the critique. Please fully address the critique.
ANSWER: Unfortunately, we are not able to see if there is something relevant to be addressed which has not been covered yet, as we responded point by point to all the comments raised by the reviewers in the previous revision. If we have missed something to complete, please let us know.
COMMENT: Minor editing of English language required
ANSWER: We have sent the manuscript to a native speaker to do the editing. We hope the changes have provided an outstanding English language for the readers.
We look forward to hearing from you soon.
